# Attacking Few-Shot Classifiers with Adversarial Support Poisoning

Elre T. Oldewage [1]   John Bronskill [1]   Richard E. Turner [1]

## Abstract

This paper examines the robustness of deployed few-shot meta-learning systems when they are fed an imperceptibly perturbed few-shot dataset, showing that the resulting predictions on test inputs can become worse than chance. This is achieved by developing a novel attack, *Adversarial Support Poisoning* or ASP, which crafts a poisoned *set* of examples. When even a small subset of malicious data points is inserted into the support set of a meta-learner, accuracy is significantly reduced. We evaluate the new attack on a variety of few-shot classification algorithms and scenarios, and propose a form of adversarial training that significantly improves robustness against both poisoning and evasion attacks.

## 1. Introduction

Standard deep learning approaches suffer from poor sample efficiency (Krizhevsky et al., 2012) which is problematic in tasks where data collection is difficult or expensive. Recently, few-shot learners have been developed which address this shortcoming by supporting rapid adaptation to a new task using only a few labeled examples (Finn et al., 2017; Snell et al., 2017).

This success has meant that few-shot learners are becoming more attractive for real-life and increasingly sensitive applications where the repercussions of confidently-wrong predictions are severe. Examples include clinical risk assessment (Sheryl Zhang et al., 2019), glaucoma diagnosis (Kim et al., 2017), identification of diseases in skin lesions (Mahajan et al., 2020), and tissue slide annotation in cancer immuno-therapy biomarker research (Lahiani et al., 2018).

As few-shot learners gain popularity, it is essential to understand how robust they are and whether there are potential avenues for their exploitation. It is well known that standard

[1]University of Cambridge. Correspondence to: Elre T. Oldewage <etv21@cam.ac.uk>.

*Accepted by the ICML 2021 workshop on A Blessing in Disguise: The Prospects and Perils of Adversarial Machine Learning.* Copyright 2021 by the author(s).

classifiers are vulnerable to inputs that have been purposefully modified in a minor way to cause incorrect predictions (Biggio & Roli, 2017). Such examples may be presented to a model either at test time, called *evasion attacks* (Biggio et al., 2017) or *adversarial examples* (Szegedy et al., 2014), or at training time, which is referred to as *poisoning* (Newsome et al., 2006; Rubinstein et al., 2009).

While previous work has considered evasion attacks or *query attacks*, in the context of few shot learners (Goldblum et al., 2019; Yin et al., 2018), data poisoning attacks have not been studied and are the focus of this paper. Data poisoning attacks are of particular relevance in the few-shot learning setting for two reasons. First, since the datasets are small, a handful of poisoned patterns might have a significant effect. Second, many applications of few-shot learning require labeled data from users to adapt the system to a new task, essentially providing a direct interface for outsiders to influence the model's behaviour.

Before detailing the key contributions of the paper, we introduce the lexicon of few-shot learning. During training, few-shot learners are typically presented with many different tasks. The model must learn to perform well on each task, hopefully arriving at a point where it can adapt effectively to a new task at test time. At test time, the model is presented with an unseen task containing a few labeled examples, the *support set*, and a number of unlabeled examples to classify, called the *query set*. The paper makes the following contributions:

1. We define a novel attack on few-shot classifiers, called *Adversarial Support Poisoning* or ASP, which applies coordinated adversarial perturbations to the support set that are calculated to minimize model accuracy over a set of query points. To the best of the authors' knowledge, this is the first work considering the impact of poisoning attacks on trained few-shot classifiers.

2. We demonstrate that few-shot classifiers are surprisingly vulnerable to our attack. ASP is more effective than the baselines considered, and generalizes well, i.e. the compromised classifier is highly likely to be inaccurate on a randomly sampled query set from the task domain.

3. We show the effectiveness of our approach against

ProtoNets (Snell et al., 2017) on the challenging META-DATASET (Triantafillou et al., 2020) benchmarks. We consider attacks against other models in the appendix.

4. We propose a form of adversarial training for few-shot learners that significantly reduces the effect of both poisoning and evasion attacks.

Section 2 provides the necessary background, Section 3 introduces the ASP attack, Section 4 presents the experimental results and Section 5 concludes the paper.

## 2. Background

We focus on image classification, denoting input images by $x \in \mathbb{R}^{ch \times W \times H}$ where $W$ is the image width, $H$ the image height, $ch$ the number of image channels and image labels $y \in \{1, \ldots, C\}$ where $C$ is the number of image classes. We use bold $\boldsymbol{x}$ and $\boldsymbol{y}$ to denote sets of images and labels.

### 2.1. Meta-Learning

We consider the few-shot image classification scenario using a meta-learning approach. Rather than a single, large dataset $D$, we assume access to a dataset $\mathcal{D} = \{\tau_t\}_{t=1}^K$ comprising a large number of training *tasks* $\tau_t$, drawn i.i.d. from a distribution $p(\tau)$. An example task is shown in Fig. A.1. The data for a task consists of a *support set* $D_S = \{(x_n, y_n)\}_{n=1}^N$ comprising $N$ elements, with the inputs $x_n$ and labels $y_n$ observed, and a *query set* $D_Q = \{(x_m^*, y_m^*)\}_{m=1}^M$ with $M$ elements for which we wish to make predictions. We may use the shorthand $D_S = \{\boldsymbol{x}, \boldsymbol{y}\}$ and $D_Q = \{\boldsymbol{x}^*, \boldsymbol{y}^*\}$. The meta-learner $g$ takes as input the support set $D_S$ and produces task-specific classifier parameters $\boldsymbol{\psi} = g(D_S)$ which are used to adapt the classifier $f$ to the current task. The classifier can now make task-specific predictions $f(x^*, \boldsymbol{\psi} = g(D_S))$ for any test input $x^* \in D_Q$. Here the inputs $x^*$ are observed and the labels $y^*$ are only observed during meta-training (i.e. training of the meta-learning algorithm). Note that the query set examples are drawn from the same set of labels as the examples in the support set. The majority of modern meta-learning methods employ *episodic* training (Vinyals et al., 2016), as detailed in A.2.1. At meta-test time, the classifier $f$ is required to make predictions for query set inputs of unseen tasks. Often, the meta-test tasks will include classes that have not been seen during meta-training, and $D_S$ will contain only a few observations.

The canonical example for modern gradient-based few-shot learning systems is MAML (Finn et al., 2017). Another widely used class of meta-learners are *amortized-inference* or *black box* based approaches e.g, VERSA (Gordon et al., 2019) and CNAPs (Requeima et al., 2019a). In these methods, the task-specific parameters $\boldsymbol{\psi}$ are generated by one or more *hyper-networks*, $g$ (Ha et al., 2016). A special case of

this approach is Prototypical Networks (ProtoNets) (Snell et al., 2017) which is based on *metric* learning and employs a nearest neighbor classifier. Our paper focuses mainly on ProtoNets, but we also consider attacks against MAML and CNAPs in the appendix.

### 2.2. Threat Model

The threat model may be summarized in terms of the adversary's goal, knowledge and capabilities. In this work, we develop poisoning attacks that degrade the model *availability* (i.e. affect prediction results indiscriminately such that it is useless) (Jagielski et al., 2018). We assume the adversary has knowledge of the model's internal workings — including gradients and other internal state information. We also assume that they can access enough data to be able to form a query set. These assumptions will be loosened in future work. We allow the attacker to manipulate some fraction of the support set and further constrain pattern modifications to be imperceptible (i.e. within some $\epsilon$ of the original image, measured using the $\ell_\infty$ norm).

## 3. Adversarial Attacks on Few-Shot Learners

Consider a few-shot learning system that has been trained and deployed as a service. A malicious party could perpetrate query or poison attacks, as described below:

**Query Attack** The attacker may want the adapted classifier to misclassify a *specific* input image. This corresponds to solving $\arg\max_\delta \mathcal{L}(f(x^* + \delta, g(\boldsymbol{x}, \boldsymbol{y})), y^*)$. Refer to Appendix A.4 for details. These kinds of attacks are essentially evasion attacks as considered in Biggio et al. (2017), which we refer to as *query attacks* in the context of few-shot learners. Many algorithms can be used to generate adversarial examples (Madry et al., 2017; Carlini & Wagner, 2017; Chen et al., 2017). Query attacks have been perpetrated successfully against few-shot learners (Goldblum et al., 2019; Yin et al., 2018), but are not the main focus of this work.

**ASP** The attacker may want the system to fail on *any* query image, as described in Section 2.2. The attacker will achieve this by computing a perturbed support set $\tilde{D}_S = \{\tilde{\boldsymbol{x}}, \boldsymbol{y}\}$ whose inputs are optimized to fool the system on a specific query set, which we call the *seed query set*, with the goal of generalizing to unseen query sets. This corresponds to solving $\arg\max_{\boldsymbol{\delta}} \mathcal{L}(f(\boldsymbol{x}^*, g(\boldsymbol{x} + \boldsymbol{\delta}, \boldsymbol{y})), \boldsymbol{y}^*)$ such that $\|\boldsymbol{\delta}\|_\infty < \epsilon$, where $D_Q = \{\boldsymbol{x}^*, \boldsymbol{y}^*\}$ denotes the seed query set and $\epsilon$ is the maximum size of the perturbation. Refer to Algorithm 1 for details. We call this novel few-shot learner attack *Adversarial Support Poisoning*, or simply ASP. Our attack is a poisoning attack, since the attacker is manipulating data that the model will use to do inference. However, it is important to note the the attack is perpetrated

at meta-test time, after the meta-learner has already been meta-trained. In contrast to a query attack, which only considers the model's behaviour on a single point at a time, ASP attacks allow the adversarial optimization function to incorporate information regarding the model's behaviour on the entire query set. In real settings, an attacker might design an attack on their own query set, hoping it will generalize to other, unseen queries. The ability to generalize may depend on $M$, the size of the seed query set. Without loss of generality, we utilize Projected Gradient Descent (PGD) (Madry et al., 2017) with an $\ell_\infty$ norm to generate perturbed support sets because it is effective, simple to implement, and can be extended to handle a set of inputs.

---

**Algorithm 1** PGD for ASP

**Require:**

$I_{min}, I_{max}$: Min/Max image intensity, $\gamma$: Step size
$L$: Number of iterations, $\epsilon$: Perturbation amount
$D_S \equiv \{\boldsymbol{x}, \boldsymbol{y}\}, D_Q \equiv \{\boldsymbol{x}^*, \boldsymbol{y}^*\}, \mathcal{L} \equiv$ cross-entropy

1: **procedure** PGDS($D_S, D_Q, f, g$)
2:     $\boldsymbol{\delta} \sim U(-\epsilon, \epsilon)$
3:     $\tilde{\boldsymbol{x}} \leftarrow \text{clip}(\boldsymbol{x} + \boldsymbol{\delta}, I_{min}, I_{max})$
4:     **for** $i \in 1, ..., L$ **do**
5:        $\boldsymbol{\delta} \leftarrow \text{sgn}(\nabla_{\tilde{\boldsymbol{x}}} \mathcal{L}(f(\boldsymbol{x}^*, g(\tilde{\boldsymbol{x}}, \boldsymbol{y})), \boldsymbol{y}^*))$
6:        $\tilde{\boldsymbol{x}} \leftarrow \text{clip}(\tilde{\boldsymbol{x}} + \gamma\boldsymbol{\delta}, I_{min}, I_{max})$
7:        $\tilde{\boldsymbol{x}} \leftarrow \boldsymbol{x} + \text{clip}(\tilde{\boldsymbol{x}} - \boldsymbol{x}, -\epsilon, \epsilon)$
8:     **end for**
9:     **return** $\tilde{\boldsymbol{x}}$
10: **end procedure**

---

## 4. Experiments

The experiments presented in the main body of the paper are carried out on ProtoNets using the challenging META-DATASET benchmark (refer to Appendix A.2 and for details on the meta-data training protocols and META-DATASET). Additional results against other models are presented in the appendix. In META-DATASET, task support sets may be large — up to 500 images across all classes. In a realistic scenario, an attacker would not likely be able to perturb all the images in such a large support set. We thus perturb only 20% of the images in the support set, which is generally considered the upper limit of manipulated patterns in conventional poisoning approaches (Jagielski et al., 2018). For all experiments, we consider the classifier's performance averaged over 500 randomly generated tasks. Each task is composed of a support set, a seed query set and up to 50 unseen query sets used for attack evaluation (some META-DATASET benchmarks do not have sufficiently many patterns available to form 50 query sets). The unseen query sets are all disjoint from the seed query set to avoid information leakage. For each task, we generate an adversarial support set using the original support set and corresponding

seed query set. The adversarial support set is then evaluated on the task's unseen query sets. We refer to the average classification accuracy on the seed query sets as the ASP *Specific* attack accuracy, and when evaluating the attack on unseen query sets we refer to it as the ASP *General* attack accuracy.

As a baseline, we also consider a *swap attack*, using the task's support set and query set to generate a query attack, then "swap" the role of the adversarial query set by using it as a support set. The swap attack is evaluated on unseen query sets.

**Attack Efficacy** Fig. 1 shows the relative decrease in accuracy of ProtoNets on META-DATASET (unnormalized results are in A.6). We compute the percentage relative decrease in classification accuracy as follows: $100\% \times (a_{clean} - a_{attack})/a_{clean}$ where $a_{clean}$ is the *clean* classification accuracy before the attack, and $a_{attack}$ is the classification accuracy after the attack. ASP *Specific* is highly effective on all datasets except MNIST, which is the easiest classification problem in the large-scale suite because there are only ten classes and the input images are simplistic. The ASP attack significantly impacts classification accuracy, easily out-performing the *Swap* baseline, in spite of the fact that only 20% of the support set shots are poisoned. Our results demonstrate that an attacker using ASP could cripple a few-shot learning system in a realistic scenario – with a challenging dataset and a limited number of perturbed patterns – far more effectively than with a simple swap attack. Results when more than 20% of the support set is poisoned are presented in Fig. A.4 of the appendix, which shows that the attack predictably becomes even stronger. The appendix also contains similar results when perpetrating attacks against CNAPs, a more sophisticated meta-learning system, in Table A.7.

**Fraction of Poisoned Patterns** We consider the importance of the number of perturbed support set points along two dimensions: specifying the fraction of classes that are adversarially perturbed and, within those classes, how many of the shots are poisoned. These and further experiments are conducted on *mini*ImageNet Vinyals et al. (2016) due to computational cost. Fig. 2 ASP shows the relative drop in 5-way classification accuracy for ProtoNets as a function of the number of poisoned classes and shots in a support set. As expected, the attack strength increases as the number of poisoned shots are increased, though perturbing even one point causes a significant drop in accuracy. The attack is stronger when the poisoned points are spread across classes, e.g. perturbing just one image from every class causes 72% accuracy drop, whereas perturbing all five shots of one class only causes 26% accuracy drop. For comparison, Fig. 2 *Swap* shows the effect of performing a swap attack with

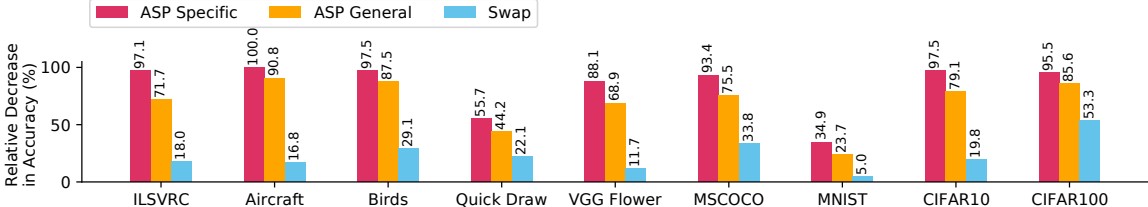

*Figure 1.* The relative drop in model accuracy for a variety of attacks using the ProtoNets algorithm on META-DATASET with $\epsilon = 0.05$, $\gamma = 0.0015$, $L = 100$, with all classes, but only 20% of the shots poisoned.

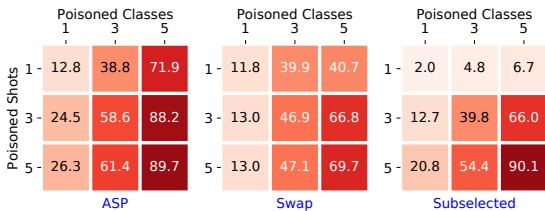

*Figure 2.* The relative drop in 5-way, 5-shot classification accuracy of ProtoNets as the number of poisoned classes shots are varied using three different methods of crafting the poisoned images. Darker colors indicate a stronger attack.

varying fractions of poisoned classes and shots. Poisoning the support set with a single malicious point, using either an adversarial support pattern or an adversarial query pattern achieves similar drops in model accuracy. However, increasing the number of poisoned shots for a particular class is significantly more effective for ASP than the swap attack because the adversarial support set's joint optimization allows for collusion among its images. We confirm this with an additional experiment shown in Fig. 2 *Subselected*, where we generate an adversarial support set with all patterns poisoned by ASP, then select subsets of these perturbed patterns to poison the original support set. Even though the ASP and *Subselected* attacks have the same fraction of poisoned patterns, ASP performs significantly better as it is jointly optimized for a specific fraction to be poisoned, whereas the patterns in *Subselected* are separated from their colluding images. We conclude that ASP is more effective than attacks that are not set-based as it enables collusion among the poisoned instances.

**Adversarial Training** To defend against attacks on few-shot classifiers, we adversarially train two different models (Goodfellow et al., 2014; Madry et al., 2017) . The first (AS) corrupts all elements of the training support sets with an ASP attack. The second (AQ) (Goldblum et al., 2019) corrupts all the elements of the training query sets with a query attack. In order to make the adversarially trained models robust against a variety of attack settings, for each task, the attacks choose $\epsilon \sim U[0.025, 0.05]$, $L \sim U[5, 100]$,

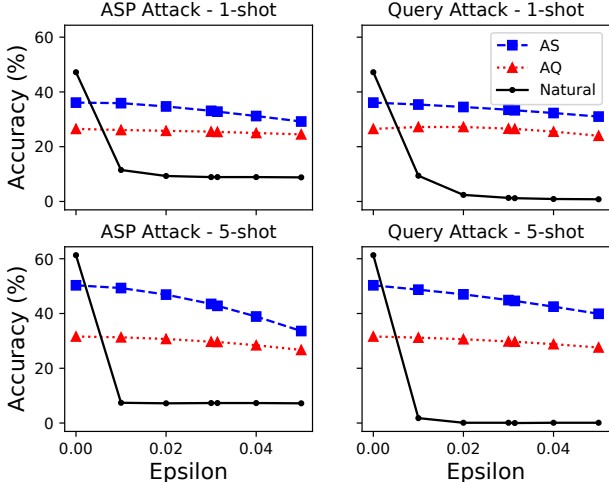

*Figure 3.* Model accuracy versus $\epsilon$ for three models: natural (no adversarial training), AS, and AQ in the MAML 5-way setting.

and $\gamma = 3\epsilon/L$. Due to the computational cost of training meta-learners, we present results for MAML in 5-way 1-shot and 5-way 5-shot scenarios. Fig. 3 and Fig. A.8 show that accuracy of models that are not adversarially trained (Natural) decreases significantly as $\epsilon$ is increased and that adversarial training (AS and AQ) mitigates the loss in accuracy to a significant extent. Interestingly, AS outperforms AQ for all models for both ASP and query attacks, even though the training was intended to mitigate ASP attacks. Both adversarially trained models are less accurate on clean data, as has been observed in Raghunathan et al. (2019), resulting in a trade-off between robustness to attacks and model performance.

## 5. Conclusions

We have introduced Adversarial Support Poisoning, an attack against trained few-shot classifiers at meta-test time. We showed that this attack is very effective, causing predictions to become worse than chance. We also devised a new form of adversarial training as a defence against ASP. Future work will consider black box and transfer ASP attacks.

ACKNOWLEDGMENTS

This work has been performed using resources provided by the Cambridge Tier-2 system operated by the University of Cambridge Research Computing Service (www.hpc.cam.ac.uk) funded by EPSRC Tier-2 capital grant EP/P020259/1.

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

# A. Appendix

The additional materials presented here are organized as follows: Appendix A.1 discusses related work; Appendix A.2 provides details regarding meta-learner training protocols and the datasets considered in the paper; Appendix A.3 summarizes the types of attacks under consideration; Appendix A.4 provides the pseudocode for a query attack against a few-shot learner and shows the algorithm generates results consistent with existing literature; Appendix A.5 provides further experimental details and tuning results for ProtoNets; Appendix A.6 provides the unnormalized results for Fig. 1 as well as additional results performing ASP attacks against CNAPs on the META-DATASET benchmark; Appendix A.7 provides additional attack results in small-scale scenarios using *mini*ImageNet against both MAML and Protonets; finally, Appendix A.8 introduces backdoor attacks and shows that ASP can also be used to perpetrate such attacks with high success rates.

## A.1. Related Work

While there has been previous work on adversarial attacks against few-shot learning systems (Goldblum et al., 2019; Yin et al., 2018), attacks that poison the support set have received little attention. Goldblum et al. (2019) devise a technique called adversarial querying (AQ) which significantly improves robustness against query attacks. They do not evaluate robustness against support set attacks at meta-test time, though they do test attacking the support set along with the query set during meta-training. Yin et al. (2018) also describe a meta-training regime to increase robustness against query attacks, but only consider the relatively weak Fast Gradient Sign Method attack (Goodfellow et al., 2014). Edmunds et al. (2017) explore the transferability of query attacks between tasks on meta-trained models and find that the attacks are indeed highly transferable, though their experiments were restricted to the MAML algorithm and the relatively simple Omniglot dataset. In this work, we focus on the effects of support set poisoning on the performance of few-shot image classifiers using a variety of learning algorithms and challenging datasets.

## A.2. Few-shot Learner Meta-training Protocols

In the following, the meta-training protocols for the various few-shot learners used in the experiments including MAML, ProtoNets, and CNAPs will be detailed.

### A.2.1. EPISODIC TRAINING

There has been an explosion of meta-learning based few-shot learning algorithms proposed in recent years. For an in-depth review see Hospedales et al. (2020). The majority of modern meta-learning methods employ *episodic* training

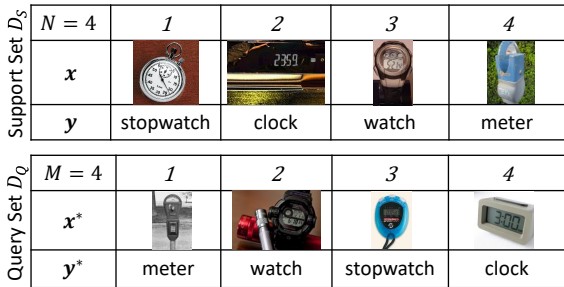

*Figure A.1.* Example task with $C = 4$ classes with $N = M = 4$.

(Vinyals et al., 2016). During meta-training, a task $\tau$ is drawn from $p(\tau)$ and randomly split into a support set $D_S$ and query set $D_Q$. Fig. A.1 depicts an example few-shot classification task. The meta-learner $g$ takes as input the support set $D_S$ and produces task-specific classifier parameters $\psi = g(D_S)$ which are used to adapt the classifier $f$ to the current task. The classifier can now make task-specific predictions $f(x^*, \psi = g(D_S))$ for any test input $x^* \in D_Q$. Refer to *Clean* in Fig. A.2. A loss function $\mathcal{L}(f(x^*, \psi), y^*)$ then computes the loss between the predictions for the label $f(x^*, \psi)$ and the true label $y^*$. Assuming that $\mathcal{L}$, $f$, and $g$ are differentiable, the meta-learning algorithm can then be trained with stochastic gradient descent by back-propagating the loss and updating the parameters of $f$ and $g$.

### A.2.2. DATASETS

***mini*ImageNet**  *mini*ImageNet is a subset of the larger Imagenet dataset (Russakovsky et al., 2015) created by Vinyals et al. (2016). It consists of 60,000 color images that is subdivided into 100 classes, each with 600 instances. The images have dimensions of $84 \times 84$ pixels. Ravi & Larochelle (2017) standardized the 64 training, 16 validation, and 20 test class splits. *mini*ImageNet has become a defacto standard dataset for benchmarking few-shot image classification methods with the following classification task configurations: (i) 5-way, 1-shot; (ii) 5-way, 5-shot.

**META-DATASET**  META-DATASET (Triantafillou et al., 2020) is composed of ten (eight train, two test) image classification datasets. We augment Meta-Dataset with three additional held-out datasets: MNIST (LeCun et al., 2010), CIFAR10 (Krizhevsky & Hinton, 2009), and CIFAR100 (Krizhevsky & Hinton, 2009). The challenge constructs few-shot learning tasks by drawing from the following distribution. First, one of the datasets is sampled uniformly; second, the "way" and "shot" are sampled randomly according to a fixed procedure; third, the classes and support / query instances are sampled. Where a hierarchical structure exists in the data (ImageNet or Omniglot), task-sampling respects the hierarchy. In the meta-test phase, the identity of the original dataset is not revealed and the tasks must be

treated independently (i.e. no information can be transferred between them). Notably, the meta-training set comprises a disjoint and dissimilar set of classes from those used for meta-test. META-DATASET is presently, the "gold standard" for evaluating few-shot classification methods. Full details are available in Triantafillou et al. (2020).

In our experiments, we excluded the Omniglot, Textures, Fungi, and Traffic Signs datasets from evaluation because their test splits are too small to allow for a fair assessment of the attack's generalization, even though the attacks reduced the classification accuracy on those datasets to approximately zero in the ASP *Specific* case.

### A.2.3. MAML META-TRAINING PROTOCOL

We meta-trained our implementation of MAML on *mini*ImageNet with identical network configuration, hyperparameters, and training protocol as prescribed in Finn et al. (2017). The meta-trained models attained the following accuracy:

*mini*ImageNet 5-way, 1-shot: $47.2 \pm 1.7\%$

*mini*ImageNet 5-way, 5-shot: $61.3 \pm 0.9\%$

### A.2.4. PROTONETS META-TRAINING PROTOCOL

We meta-trained our implementation of ProtoNets on *mini*ImageNet with identical network configuration, hyperparameters, and training protocol as prescribed in Snell et al. (2017). The meta-trained models attained the following accuracy:

*mini*ImageNet 5-way, 1-shot: $46.8 \pm 0.6\%$

*mini*ImageNet 5-way, 5-shot: $65.1 \pm 0.5\%$

For META-DATASET, we meta-trained ProtoNets using the code from Requeima et al. (2019b) with FiLM feature adaptation. We made modifications to the code to enable various adversarial attacks. The meta-trained model attained the following results:

ilsvrc 2012: $55.1 \pm 1.1$, omniglot: $90.8 \pm 0.6$, aircraft: $82.3 \pm 0.6$, cu birds: $74.0 \pm 0.9$, dtd: $63.4 \pm 0.7$, quickdraw: $75.3 \pm 0.8$, fungi: $44.6 \pm 1.0$, vgg flower: $90.3 \pm 0.5$, traffic sign: $67.9 \pm 0.8$, mscoco: $40.8 \pm 1.0$, mnist: $91.4 \pm 0.5$, cifar10: $72.5 \pm 0.8$, cifar100: $58.4 \pm 1.0$.

### A.2.5. CNAPS META-TRAINING PROTOCOL

For all the CNAPS experiments, we use the code provided by the the CNAPS authors (Requeima et al., 2019b). We made modifications to the code to enable various adversarial attacks and used FiLM feature adaptation. We follow an identical dataset configuration and meta-training process as prescribed in Requeima et al. (2019b).

### A.3. Attack Summary

We summarize the types of attacks we perform in Fig. A.2. The first scenario, *Clean*, illustrates how the meta-learner $g$ performs a test-time task, taking the support set $D_S$ as input to produce parameters $\psi = g(D_S)$ which are used to adapt the classifier $f$ to the task. The classifier makes task-specific predictions $f(x^*, \psi = g(D_S))$ for any test input $x^* \in D_Q$. *ASP* and *Query* illustrate the ASP and Query attacks as discussed in Section 3 and Appendix A.4. *Swap* illustrates a swap attack, which is used as a baseline comparison for ASP, where a set of images are perturbed with a query attack and then inserted into the support set. Query attacks are typically cheaper to compute, since they do not require back-propagation through the meta-learner, so it is an important baseline to consider. *Label Shift* is a simple attack on the support set which involves mislabelling the support set images by shifting the true label index by one in a modulo arithmetic fashion. We consider systematic mislabeling in this way to be a strong attack for comparison, though this "attack" would be easily detected by inspection. We thus only consider the label shift baseline for the small scale results presented in Appendix A.7.

### A.4. Query Attacks

We present our algorithm for performing query attacks with PGD in Algorithm A.1. Using this algorithm, we attack MAML and ProtoNets with settings that match the experiments by (Goldblum et al., 2019) to ensure that our attack performs approximately the same, as expected. The attack settings used are $L = 20$, $\epsilon = \frac{8}{255}$, $\gamma = \frac{2}{255}$, with an untargeted loss function. Our results are shown in Table A.1 and the relevant results from (Goldblum et al., 2019) are shown in Table A.2. Our models perform similarly when presented with clean data and when attacked using PGD, as expected.

*Table A.1.* The classification accuracy (%) when performing our query attack against MAML and ProtoNets models in the 5-way, 1-shot and 5-shot configurations on *mini*ImageNet. PGD settings were $L$=20, with $\epsilon = \frac{8}{255}$, $\gamma = \frac{2}{255}$. All figures are percentages and the $\pm$ sign indicates the 95% confidence interval over tasks.

|  | MAML | | ProtoNets | |
|---|---|---|---|---|
|  | Clean | Adversarial | Clean | Adversarial |
| 5-way, 1 shot | $47.0 \pm 0.3$ | $0.0 \pm 0.0$ | $46.6 \pm 0.3$ | $0.0 \pm 0.0$ |
| 5-way, 5 shot | $60.7 \pm 0.1$ | $0.0 \pm 0.0$ | $64.7 \pm 0.1$ | $0.0 \pm 0.0$ |

### A.5. Further Experimental Details

In our experiments, all the input images were re-scaled to have pixel values between $-1$ and $1$. We considered perturbations using the $\ell_\infty$ norm, on a scale of $[-1, 1]$, so that $\epsilon = 0.1$ corresponds to allowing $\pm 10\%$ or an absolute change of $\pm 0.2$ to the intensity of each pixel in an image.

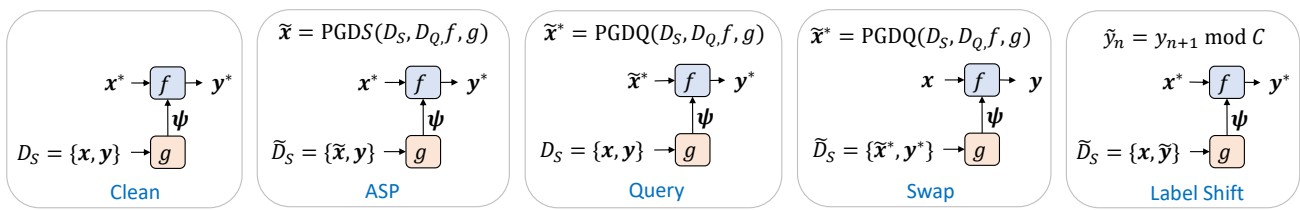

*Figure A.2.* Attacks on meta-learning based few-shot image classifiers. $f$ and $g$ denote the classifier and trained meta-learner, respectively. Each diagram depicts how an attack is applied and includes an expression for the attack's computation using Algorithms 1 and A.1.

---

**Algorithm A.1** PGD for Query Attack

**Require:**

    $I_{min}$: Minimum image intensity

    $I_{max}$: Maximum image intensity

    $L$: Number of iterations

    $\epsilon$: Perturbation amount

    $\gamma$: Step size

    $D_S \equiv \{\boldsymbol{x}, \boldsymbol{y}\}$

    $D_Q \equiv \{\boldsymbol{x}^*, \boldsymbol{y}^*\}$

    ▷ We use cross-entropy loss for $\mathcal{L}$.

1: **procedure** PGDQ($D_S, D_Q, f, g$)
2:     $\boldsymbol{\delta} \sim U(-\epsilon, \epsilon)$
3:     $\tilde{\boldsymbol{x}}^* \leftarrow \text{clip}(\boldsymbol{x}^* + \boldsymbol{\delta}, I_{min}, I_{max})$
4:     **for** $n \in 1, ..., L$ **do**
5:         $\boldsymbol{\delta} \leftarrow \text{sgn}(\nabla_{\tilde{\boldsymbol{x}}^*} \mathcal{L}(f(\tilde{x}^*, g(x, y)), y^*))$
6:         $\tilde{\boldsymbol{x}}^* \leftarrow \text{clip}(\tilde{\boldsymbol{x}}^* + \gamma\boldsymbol{\delta}, I_{min}, I_{max})$
7:         $\tilde{\boldsymbol{x}}^* \leftarrow x + \text{clip}(\tilde{\boldsymbol{x}}^* - \boldsymbol{x}^*, -\epsilon, \epsilon)$
8:     **end for**
9:     **return** $\tilde{\boldsymbol{x}}^*$
10: **end procedure**

---

*Table A.2.* Results reproduced from Goldblum et al. (2019) where possible. The table shows classification accuracy (%) when performing a query attack against MAML and ProtoNets models in the 5-way, 1-shot and 5-shot configurations on *mini*ImageNet. Results were tested on 150000 samples. PGD settings were $L=20$, with $\epsilon = \frac{8}{255}$, $\gamma = \frac{2}{255}$. All figures are percentages.

| | MAML | | ProtoNets | |
|---|---|---|---|---|
| | Clean | Adversarial | Clean | Adversarial |
| 5-way, 1 shot | 45.04 | 0.03 | 43.26 | 0.00 |
| 5-way, 5 shot | 60.25 | 0.03 | 70.23 | 0.00 |

We calculated the perturbation step size $\gamma$ to depend on $\epsilon$ and the maximum number of iterations, so that $\gamma = r\frac{\epsilon}{L}$, where $r$ is a scaling coefficient. We observed that the optimal values for $r$ depend on the numbers of shots, and varies with $\epsilon$ and $L$.

Our experiments here performed both targeted and untargeted attacks. When perpetrating ASP attacks, we consider

two variations of the loss function: *all*, in which the entire query set is used; and *single*, in which a single, random point in the query set is chosen for the loss calculation at each attack iteration, with the intention that the additional stochasticity may prevent the attack from getting stuck in local optima. We found that the *all* strategy performed best when combined with a targeted attack, whereas the *single* strategy worked better in combination with an untargeted attack. Unless otherwise specified, we use the targeted, *all* loss strategy. The ASP results presented in the main body of the paper are all targeted attacks using the *all* loss strategy.

Example images from an ASP attack are shown in Fig. A.3.

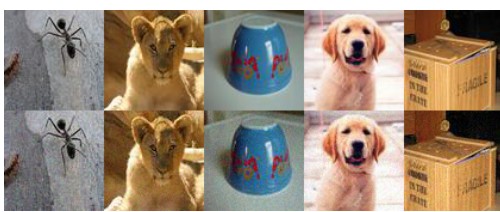

*Figure A.3.* Pairs of images from the *mini*ImageNet dataset where the top is unperturbed, while the bottom is adversarially perturbed by a PGD ASP attack with $\epsilon = 0.05$, $\gamma = 0.0015$, and $L = 100$.

**Protonets Tuning** The results of our tuning experiments are provided for ProtoNets on *mini*ImageNet in Table A.3 - Table A.5. In general, larger values of $r$ (i.e. larger step sizes) performed better as the number of PGD iterations increased. Although the *single* loss strategy did not perform well for 1-shot classification, its performance increased at higher shots, often out-performing the *all* strategy for sufficiently large numbers of PGD iterations, even though the *Specific* accuracy did not go to $0.0\%$. The *all* strategy performed significantly better than *single* when $L$ was low.

**A.6. Additional Large-Scale Attack Results**

Here, we present the unnormalized numbers for Fig. 1 in Table A.6. We also present results for a similar attack, perpetrated against CNAPS in Table A.7, showing that our attack is also effective against CNAPS in a large-scale scenario. Fig. A.4 shows the effectiveness of both ASP and swap attacks increase as the fraction of poisoned shots within a

*Table A.3.* Accuracy of ProtoNets 5-way 1-shot, with perturbation size $\epsilon = 0.05$ when varying the loss function (targeted *all* or untargeted *single*), PGD iterations (given in the column headers) and step size ratio ($r$), over 100 tasks. Seed query set size is fixed at $13N$. Clean accuracy is $47.5 \pm 2.0\%$. All figures are percentages and the $\pm$ sign indicates the 95% confidence interval over tasks. Bold text indicates the lowest score.

|  | | 20 | | 50 | | 100 | | 200 | | 500 | |
|---|---|---|---|---|---|---|---|---|---|---|---|
|  | $r$ | Specific | General | Specific | General | Specific | General | Specific | General | Specific | General |
| all | 0.25 | 7.4±0.9 | 17.0±0.5 | 7.0±0.9 | 16.8±0.5 | 6.9±1.0 | 16.7±0.5 | 6.8±0.9 | 16.8±0.5 | 6.9±0.9 | 16.6±0.5 |
|  | 0.5 | **3.4±0.6** | 13.2±0.4 | 1.5±0.3 | 12.2±0.4 | 1.2±0.3 | 11.8±0.4 | 1.3±0.3 | 11.8±0.4 | 1.3±0.3 | 11.8±0.4 |
|  | 1 | 3.7±0.6 | **12.9±0.4** | 0.8±0.2 | 10.9±0.4 | 0.3±0.2 | 10.1±0.4 | 0.1±0.1 | 9.9±0.4 | 0.1±0.1 | 10.1±0.4 |
|  | 1.5 | 3.9±0.7 | **12.9±0.4** | 0.9±0.3 | **10.8±0.4** | **0.1±0.1** | 9.6±0.4 | **0.0±0.0** | 9.3±0.4 | **0.0±0.0** | 9.6±0.4 |
|  | 2 | 4.9±0.8 | 13.5±0.4 | 1.2±0.3 | **10.8±0.4** | 0.2±0.1 | 9.4±0.4 | **0.0±0.0** | 9.2±0.4 | **0.0±0.0** | 9.3±0.4 |
|  | 3 | 7.1±1.1 | 14.4±0.4 | 1.5±0.4 | 11.0±0.4 | 0.2±0.1 | **9.3±0.4** | **0.0±0.0** | 9.0±0.4 | **0.0±0.0** | 8.8±0.3 |
| single | 0.25 | 32.8±2.1 | 34.0±0.6 | 29.9±1.9 | 32.6±0.5 | 30.6±2.0 | 33.8±0.6 | 30.6±1.9 | 34.1±0.6 | 31.3±2.0 | 35.0±0.6 |
|  | 0.5 | 26.1±1.8 | 27.7±0.5 | 21.6±1.6 | 25.6±0.5 | 21.2±1.7 | 25.0±0.5 | 20.9±1.7 | 25.5±0.5 | 21.2±1.7 | 26.1±0.5 |
|  | 1 | 21.3±1.6 | 23.2±0.5 | 15.0±1.5 | 18.8±0.5 | 13.5±1.3 | 17.4±0.4 | 12.6±1.3 | 17.7±0.5 | 12.0±1.4 | 17.4±0.5 |
|  | 1.5 | 20.3±1.7 | 22.1±0.5 | 13.1±1.5 | 16.4±0.4 | 10.9±1.2 | 14.7±0.4 | 9.2±1.1 | 14.3±0.4 | 8.2±1.1 | 13.6±0.4 |
|  | 2 | **19.5±1.6** | **21.7±0.5** | 12.6±1.4 | 15.4±0.4 | 10.0±1.2 | 13.2±0.4 | 7.8±1.0 | 12.3±0.4 | 6.4±1.0 | 11.9±0.4 |
|  | 3 | 20.8±1.8 | 22.4±0.5 | **11.9±1.3** | **14.9±0.4** | **8.9±1.1** | **12.0±0.4** | **6.4±1.0** | **10.7±0.4** | **5.0±0.8** | **9.6±0.4** |

*Table A.4.* Accuracy of ProtoNets 5-way 5-shot, with perturbation size $\epsilon = 0.05$ when varying the loss function (targeted *all* or untargeted *single*), PGD iterations (given in the column headers) and step size ratio ($r$), over 100 tasks. Seed query set size is fixed at $7N$. Clean accuracy is $64.2 \pm 1.6\%$. All figures are percentages and the $\pm$ sign indicates the 95% confidence interval over tasks. Bold text indicates the lowest score.

|  | | 20 | | 50 | | 100 | | 200 | | 500 | |
|---|---|---|---|---|---|---|---|---|---|---|---|
|  | $r$ | Specific | General | Specific | General | Specific | General | Specific | General | Specific | General |
| all | 0.25 | **1.1±0.2** | 9.8±0.2 | 0.1±0.0 | 9.0±0.2 | **0.0±0.0** | 8.7±0.2 | **0.0±0.0** | 8.7±0.2 | **0.0±0.0** | 8.7±0.2 |
|  | 0.5 | 1.5±0.3 | 10.3±0.2 | **0.0±0.0** | 8.1±0.2 | **0.0±0.0** | 7.8±0.2 | **0.0±0.0** | 7.6±0.2 | **0.0±0.0** | 7.7±0.2 |
|  | 1 | 1.9±0.5 | 9.8±0.2 | 0.1±0.0 | **8.0±0.2** | **0.0±0.0** | 7.4±0.2 | **0.0±0.0** | 6.9±0.1 | **0.0±0.0** | 7.0±0.1 |
|  | 1.5 | 2.3±0.5 | **9.7±0.2** | 0.1±0.1 | 8.1±0.2 | **0.0±0.0** | 7.1±0.2 | **0.0±0.0** | 6.8±0.1 | **0.0±0.0** | 6.6±0.1 |
|  | 2 | 3.3±0.6 | 10.6±0.2 | 0.1±0.1 | 8.2±0.2 | **0.0±0.0** | 7.2±0.2 | **0.0±0.0** | 6.6±0.1 | **0.0±0.0** | 6.5±0.1 |
|  | 3 | 6.4±0.9 | 11.9±0.2 | 0.2±0.1 | 8.1±0.2 | **0.0±0.0** | 7.3±0.2 | **0.0±0.0** | 6.5±0.1 | **0.0±0.0** | 6.3±0.1 |
| single | 0.25 | 36.0±1.6 | 38.6±0.3 | 35.9±1.7 | 39.3±0.3 | 34.9±1.7 | 39.3±0.3 | 33.5±1.9 | 38.7±0.3 | 33.5±1.9 | 39.3±0.4 |
|  | 0.5 | 24.7±1.4 | 26.3±0.3 | 17.3±1.4 | 20.7±0.3 | 14.6±1.2 | 18.4±0.3 | 12.2±1.2 | 17.0±0.3 | 10.8±1.2 | 16.6±0.3 |
|  | 1 | 18.3±1.4 | 19.7±0.3 | 9.1±1.1 | 11.2±0.2 | 5.9±0.7 | 8.2±0.2 | 3.7±0.5 | 6.5±0.2 | 2.9±0.5 | 6.1±0.2 |
|  | 1.5 | **17.9±1.4** | **19.0±0.3** | 7.7±1.0 | 9.5±0.2 | 4.5±0.6 | 6.2±0.2 | 2.4±0.3 | 4.4±0.1 | 1.6±0.3 | 3.9±0.1 |
|  | 2 | 18.5±1.4 | 19.4±0.3 | **7.5±1.0** | **9.1±0.2** | **4.0±0.6** | 5.5±0.1 | **2.0±0.3** | 3.7±0.1 | 1.2±0.2 | 3.1±0.1 |
|  | 3 | 19.8±1.4 | 21.3±0.3 | 8.0±1.0 | 9.6±0.2 | 4.1±0.6 | **5.4±0.1** | **2.0±0.3** | **3.3±0.1** | **1.1±0.2** | **2.6±0.1** |

*Table A.5.* Accuracy of ProtoNets 5-way 10-shot, with perturbation size $\epsilon = 0.05$ when varying the loss function (targeted *all* or untargeted *single*), PGD iterations (given in the column headers) and step size ratio ($r$), over 100 tasks. Seed query set size is fixed at $6N$. Clean accuracy is $71.7 \pm 1.1\%$. All figures are percentages and the $\pm$ sign indicates the 95% confidence interval over tasks. Bold text indicates the lowest score.

| | | 20 | | 50 | | 100 | | 200 | | 500 | |
|---|---|---|---|---|---|---|---|---|---|---|---|
| | $r$ | Specific | General | Specific | General | Specific | General | Specific | General | Specific | General |
| all | 0.25 | 1.7±0.2 | **8.5±0.1** | 0.4±0.1 | 7.6±0.1 | 0.2±0.1 | 7.5±0.1 | 0.2±0.1 | 7.5±0.1 | 0.2±0.1 | 7.5±0.1 |
| | 0.5 | 2.0±0.3 | **8.5±0.1** | 0.2±0.1 | 7.2±0.1 | 0.0±0.0 | 6.9±0.1 | 0.0±0.0 | 6.9±0.1 | 0.0±0.0 | 7.0±0.1 |
| | 1 | 2.3±0.4 | **8.5±0.1** | 0.2±0.1 | **7.1±0.1** | 0.0±0.0 | 6.5±0.1 | 0.0±0.0 | 6.4±0.1 | 0.0±0.0 | 6.5±0.1 |
| | 1.5 | 2.9±0.5 | 8.9±0.2 | 0.2±0.1 | **7.1±0.1** | 0.0±0.0 | 6.7±0.1 | 0.0±0.0 | 6.2±0.1 | 0.0±0.0 | 6.2±0.1 |
| | 2 | 4.6±0.6 | 10.1±0.2 | 0.4±0.1 | 7.2±0.1 | 0.0±0.0 | **6.4±0.1** | 0.0±0.0 | **6.0±0.1** | 0.0±0.0 | 6.0±0.1 |
| | 3 | 8.0±1.0 | 12.5±0.2 | 0.7±0.2 | 7.3±0.1 | 0.1±0.0 | 6.5±0.1 | 0.0±0.0 | 6.1±0.1 | 0.0±0.0 | **5.9±0.1** |
| single | 0.25 | 23.6±1.4 | 24.3±0.2 | 16.5±1.2 | 18.2±0.2 | 13.4±1.2 | 15.6±0.2 | 11.2±1.1 | 14.1±0.2 | 11.8±1.0 | 15.1±0.2 |
| | 0.5 | 17.3±1.5 | 17.9±0.2 | 8.3±0.9 | 9.3±0.2 | 5.6±0.7 | 6.7±0.1 | 3.8±0.5 | 5.2±0.1 | 3.0±0.4 | 4.8±0.1 |
| | 1 | 15.8±1.4 | **16.5±0.2** | 6.3±0.7 | **7.1±0.1** | 3.7±0.5 | 4.6±0.1 | 2.3±0.3 | 3.2±0.1 | 1.6±0.2 | 2.6±0.1 |
| | 1.5 | 16.6±1.3 | 17.0±0.2 | 6.4±0.7 | 7.2±0.1 | 3.5±0.5 | **4.4±0.1** | 2.1±0.3 | 2.9±0.1 | 1.4±0.2 | 2.2±0.1 |
| | 2 | 17.9±1.3 | 18.3±0.2 | 7.0±0.8 | 7.7±0.1 | 3.8±0.5 | 4.6±0.1 | 2.1±0.3 | **2.8±0.1** | 1.3±0.2 | 2.1±0.1 |
| | 3 | 19.0±1.2 | 19.7±0.2 | 7.8±0.8 | 8.4±0.2 | 4.2±0.6 | 5.1±0.1 | 2.2±0.3 | 2.9±0.1 | 1.4±0.2 | **2.0±0.1** |

*Table A.6.* Accuracy of ProtoNets with FiLM on the META-DATASET benchmark in the *Clean*, *Specific* and *General* scenarios when attacking with an adversarial support set, with $\epsilon = 0.05$, $\gamma = 0.0015$, $L = 100$, with all classes, but only 20% of the shots poisoned. All figures are percentages and the $\pm$ sign indicates the 95% confidence interval over 500 tasks.

| | Clean | Specific | General | Swap |
|---|---|---|---|---|
| ilsvrc_2012 | 52.2±0.2 | 1.5±0.1 | 14.8±0.1 | 42.8±0.2 |
| aircraft | 78.5±0.5 | 0.0±0.0 | 7.2±0.2 | 65.3±1.0 |
| cu_birds | 71.4±1.1 | 1.8±0.2 | 8.9±0.4 | 50.6±2.0 |
| quickdraw | 74.1±0.1 | 32.8±1.2 | 41.3±0.2 | 57.7±0.3 |
| vgg_flower | 89.1±0.6 | 10.6±0.8 | 27.7±1.0 | 78.7±1.8 |
| traffic_sign | 35.2±0.4 | 4.1±0.3 | 12.3±0.3 | 24.8±0.5 |
| mscoco | 44.1±0.3 | 2.9±0.1 | 10.8±0.1 | 29.2±0.3 |
| mnist | 90.3±0.1 | 58.8±1.4 | 68.9±0.2 | 85.8±0.2 |
| cifar10 | 64.6±0.1 | 1.6±0.1 | 13.5±0.1 | 51.8±0.2 |
| cifar100 | 53.5±0.6 | 2.4±0.1 | 7.7±0.2 | 25.0±0.8 |

*Table A.7.* Accuracy of CNAPs on the META-DATASET benchmark in the *Clean*, *Specific* and *General* scenarios when attacking with an adversarial support set, with $\epsilon = 0.05$, $\gamma = 0.0015$, $L = 100$, with all classes, but only 20% of the shots poisoned. All figures are percentages and the $\pm$ sign indicates the 95% confidence interval over 500 tasks.

| | Clean | Specific | General | Swap |
|---|---|---|---|---|
| ilsvrc_2012 | 46.7±0.2 | 0.2±0.0 | 10.5±0.1 | 37.6±0.2 |
| aircraft | 70.3±0.6 | 0.0±0.0 | 9.4±0.2 | 61.9±0.9 |
| cu_birds | 68.4±0.9 | 0.1±0.0 | 6.5±0.3 | 56.9±1.7 |
| quickdraw | 69.1±0.1 | 2.1±0.1 | 12.2±0.1 | 53.8±0.3 |
| vgg_flower | 83.5±0.7 | 1.5±0.2 | 16.0±0.6 | 72.7±1.7 |
| traffic_sign | 32.8±0.4 | 0.4±0.1 | 9.7±0.2 | 27.1±0.4 |
| mscoco | 43.4±0.3 | 0.4±0.0 | 9.4±0.1 | 31.8±0.3 |
| mnist | 88.9±0.1 | 12.1±0.7 | 32.7±0.1 | 84.4±0.2 |
| cifar10 | 60.7±0.1 | 0.1±0.0 | 12.9±0.1 | 53.6±0.2 |
| cifar100 | 47.3±0.6 | 0.4±0.0 | 6.5±0.2 | 30.1±0.8 |

class increases. It further demonstrates the superiority of ASP compared to the swap attack in the large-scale few-shot learning setting.

## A.7. Additional Small-Scale Experiments

**Small-Scale Results** Fig. A.5 depicts the relative decrease in 5-way classification accuracy due to a variety of attacks for two values of $\epsilon$ on *mini*ImageNet. As in the main body of the paper, we compute the percentage relative decrease in classification accuracy as follows: $100\% \times (a_{clean} - a_{attack})/a_{clean}$ where $a_{clean}$ is the *clean* classification accuracy before the attack, and $a_{attack}$ is the classification accuracy after the attack. We include two dif-

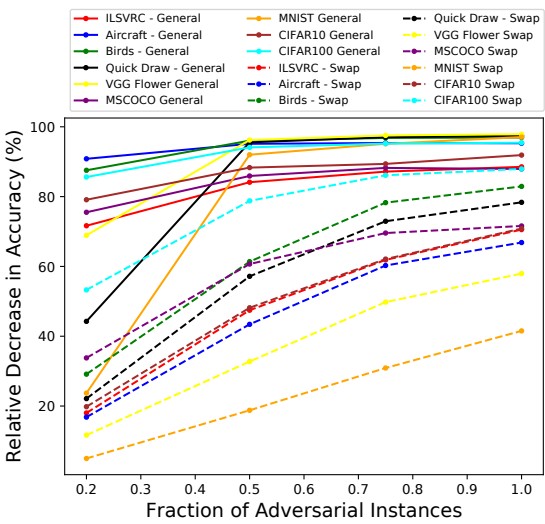

*Figure A.4.* Relative drop in model accuracy for ASP (solid) and swap (dashed) versus the fraction of poisoned shots for ProtoNets with $\epsilon = 0.05$, $\gamma = 0.0015$, $L = 100$, with all classes affected.

ferent results for the ASP *General* scenario: ASP *General (1x)*, for which the seed query set size is the same size as the support set (i.e. $M = N$) and ASP *General (10x)*, where $M = 10N$.

The ASP *Specific* attack is $100\%$ effective in all cases, indicating that the adversarial optimization is acting as desired. The attack's generalization to unseen query sets, as measured by the ASP *General (x10)* attack, successfully beats all the baselines. The ASP *General (x1)* attack, is noticeably weaker but still beats the *Swap* baseline on Protonets and the *Label Shift* baseline on MAML. Since we are measuring generalization in these scenarios, it is expected that an attack with less data to learn from does not generalize as well. We note that the results presented in the main body of the paper, on more difficult and realistic problems, do not require large seed query sets to succeed. The *Label Shift* attack causes a large drop in accuracy, but is easily detected by inspection, and so does not provide like-for-like comparison. Even so, ASP *General (10x)* is more effective than *Label Shift* in all scenarios. From Fig. A.5, there does not appear to be a difference between ProtoNets and MAML in terms of robustness to the ASP attack, though MAML is significantly more vulnerable to the swap attack. For 5-shot classification, increasing the perturbation size does not consistently increase the attack's impact. This is because the step size $\gamma$ was tuned for the 1-shot case, but was not optimal for the 5-shot problem. For consistency, we used the same PGD settings for both in Fig. A.5. Since we are primarily interested in degrading model accuracy for unseen inputs, we consider only the ASP *General* case for the remainder of the results, unless otherwise specified.

To supplement Fig. A.5, we provide the unnormalized in Tables A.8 and A.9, for the 1-shot and 5-shot scenarios, respectively. All adversarial query points used in the swap attacks achieved approximately $100\%$ fooling rates when presented to the learner as query attacks.

*Table A.8.* The classification accuracy for a variety of attacks against MAML and ProtoNets models in the 5-way, 1-shot *mini*ImageNet configuration, with $M = 20N$. All support images were perturbed. PGD settings were $L$=100, with $\gamma = 0.0015$ for $\epsilon = 0.05$, and $\gamma = 9.4e{-}4$ for $\epsilon = 0.0314$. All figures are percentages and the $\pm$ sign indicates the 95% confidence interval over tasks.

| | $\epsilon$ | Label Shift | Noise | ASP Specific | ASP General | Swap |
|---|---|---|---|---|---|---|
| Protonets | 0.0314 | 13.0±0.2 | 46.3±0.3 | 1.3±0.1 | 9.8±0.2 | 19.2±0.2 |
| (Clean: 47.5±0.3) | 0.05 | 13.0±0.2 | 43.8±0.3 | 1.1±0.1 | 9.4±0.2 | 18.2±0.2 |
| MAML | 0.0314 | 20.6±0.2 | 46.9±0.3 | 0.8±0.1 | 8.9±0.2 | 12.0±0.2 |
| (Clean: 46.9±0.3) | 0.05 | 20.6±0.2 | 46.2±0.3 | 0.7±0.1 | 8.8±0.2 | 11.2±0.2 |

*Table A.9.* The classification accuracy (%) for a variety of attacks against MAML and ProtoNets models in the 5-way, 5-shot *mini*ImageNet configuration, with $M = 20N$. All support images were perturbed. PGD settings were $L$=100, with $\gamma = 0.0015$ for $\epsilon = 0.05$, and $\gamma = 9.4e{-}4$ for $\epsilon = 0.0314$. All figures are percentages and the $\pm$ sign indicates the 95% confidence interval over tasks.

| | $\epsilon$ | Label Shift | Noise | ASP Specific | ASP General | Swap |
|---|---|---|---|---|---|---|
| Protonets | 0.0314 | 8.9±0.1 | 64.2±0.1 | 0.6±0.1 | 6.2±0.1 | 19.4±0.1 |
| (Clean: 64.6±0.1) | 0.05 | 8.9±0.1 | 58.4±0.1 | 0.9±0.1 | 6.4±0.1 | 19.5±0.1 |
| MAML | 0.0314 | 19.6±0.1 | 60.6±0.1 | 1.2±0.1 | 7.2±0.1 | 9.1±0.1 |
| (Clean: 61.4±0.1) | 0.05 | 19.6±0.1 | 60.0±0.1 | 1.5±0.1 | 7.3±0.1 | 9.3±0.1 |

**Fraction of Poisoned Patterns** In addition to Fig. 2 in Section 4 of the paper, we provide similar plots for MAML, ProtoNets and CNAPs on 5-way, 1-shot and 5-way, 5-shot problems in Figures A.6 and A.7. Note that for these results, MAML and ProtoNets were performing classification on *mini*ImageNet, whereas CNAPs was performing classification on ILSVRC-2012, which is a more difficult problem.

**Adversarial Training for Protonets** Fig. A.8 is the same as Fig. 3 except that the ProtoNets algorithm is used instead of MAML. The overall trend is consistent between MAML and ProtoNets, though adversarial training is more robust in the MAML case as the AS and AQ accuracy decays to a lesser extent as $\epsilon$ is increased. In addition, the difference between AS and AQ accuracy is smaller in the ProtoNets case .

## A.8. Backdoor Poisoning Attacks

We also consider targeted backdoor poisoning attacks, where the goal is for a specific input $x^*$ in the query set

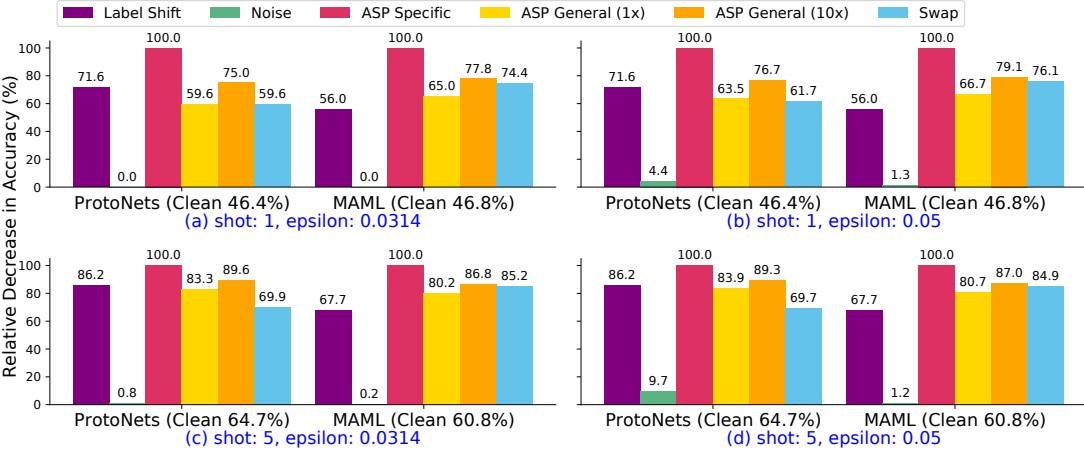

*Figure A.5.* The relative drop in classification accuracy for a variety of attacks against MAML and ProtoNets models in the 5-way *mini*ImageNet configuration. For ASP General (1x), $M=N$, and for ASP General (10x), $M=10N$. All support images were perturbed. PGD settings were $L=100$, with $\gamma = 0.0015$ for $\epsilon = 0.05$, and $\gamma = 9.4e{-}4$ for $\epsilon = 0.0314$.

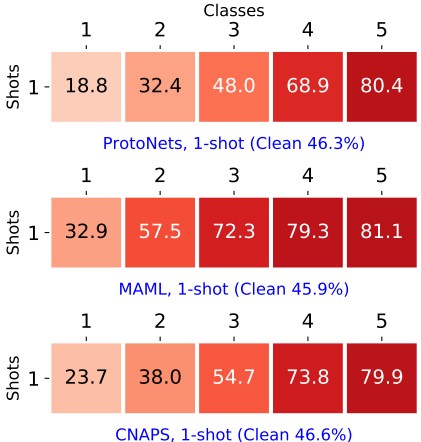

*Figure A.6.* The relative drop in 5-way, 1-shot classification accuracy of ProtoNets, MAML and CNAPs as the number of poisoned classes and poisoned shots within those classes are varied when performing ASP attacks. Darker colors indicate a stronger attack. Attacks were calculated with $\epsilon = 0.05$, $\gamma = 0.0015$, $L = 200$, $M = 13N$, averaged over 250 tasks.

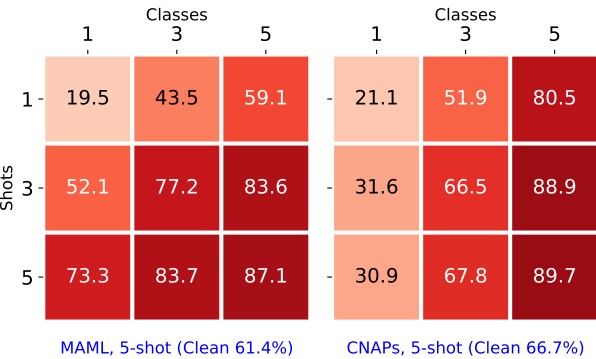

*Figure A.7.* The relative drop in 5-way, 5-shot classification accuracy of MAML and CNAPs as the number of poisoned classes and poisoned shots within those classes are varied when performing ASP attacks. Darker colors indicate a stronger attack. Attacks were calculated with $\epsilon = 0.05$, $\gamma = 0.0015$, $L = 200$, $M = 7N$, averaged over 250 tasks. The ProtoNets, 5-shot scenario can be found in the main body of the paper.

to be misclassified as class $t$. We assume the attacker can only poison one element, $\tilde{x}$, in the support set. The support set point $x$ that is used to generate $\tilde{x}$ is chosen from the target class (i.e. $y = t$). An attacker has a specific $x^*$ and $t$ in mind, but to evaluate the attack's feasibility, we randomly choose $x$, $x^*$ and $t$ such that $y = t$, and so that $x^*$ is initially correctly classified (this removes the case where the classifier assigns the incorrect label to $x^*$, which may inflate our results). The poisoned instance is generated as usual, except that the loss is only calculated with respect to $x^*$, not with respect to the entire seed seed query set, as for

the usual ASP attack. We evaluate the backdoor attack in the *Specific* scenario by presenting the model with both the support set and query set used in attack generation (where the support set contains $\tilde{x}$ instead of $x$). We also evaluate the attack's ability to generalize (i.e. the *General* scenario) by inserting $\tilde{x}$ into a different support set that is randomly generated for the same task. We call the backdoor attack a *success* in either the *Specific* or *General* scenario if the targeted instance, $x^*$ is classified as belonging to class $t$ at meta-test time. Table A.10 shows that ASP backdoor attacks are very successful against the ProtoNets *Natural* model. In the 1-shot case, ASP is able to cause the targeted point to be misclassified as class $t$ more than 99% of the

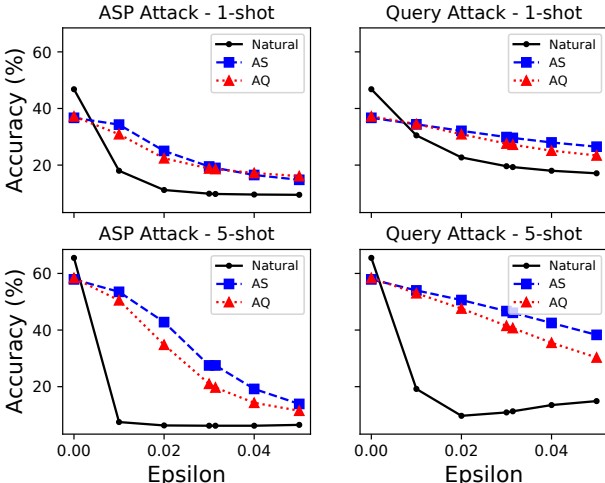

*Figure A.8.* Model accuracy versus $\epsilon$ for three models: natural (no adversarial training), AS, and AQ in the ProtoNets 5-way setting.

*Table A.10.* Backdoor attack against naturally and adversarially trained (ASP, AQ) 5-way ProtoNets models. *Success* indicates the % of the targeted pattern's labels that were flipped to $t$.

| | Model | Overall Accuracy | | Success | Success |
|---|---|---|---|---|---|
| | | Before | After | Specific (%) | General (%) |
| 1-shot | Natural | 51.1±1.7 | 30.2±1.6 | 99.6±0.5 | 99.5±0.1 |
| | AQ | 43.3±1.6 | 24.5±1.5 | 98.6±1.0 | 98.7±0.1 |
| | AS | 41.0±1.5 | 23.9±1.0 | 99.8±0.4 | 99.9±0.0 |
| 5-shot | Natural | 64.3±0.9 | 58.7±1.0 | 93.0±2.2 | 85.4±0.4 |
| | AQ | 57.5±1.0 | 56.0±1.0 | 29.6±4.0 | 30.1±0.6 |
| | AS | 56.6±1.0 | 55.0±1.0 | 40.6±4.3 | 37.0±0.6 |

time, and more than 85% in the 5-shot setting. We conclude that that an ASP backdoor attack is highly likely to succeed, even in the general case, using just one poisoned example inserted into an unknown support set. For backdoor attacks to remain undetected, the accuracy of the model should be preserved (except for the special, targeted point). Table A.10 demonstrates that the overall accuracy of the model did not decrease significantly with the introduction of the poisoned input in the 5-shot setting. For 1-shot, the accuracy drops quite significantly because the now incorrect, targeted point forms 20% of the query set, but the accuracy never drops by more than 20%. The ASP backdoor attack is thus difficult to detect by examining model accuracy.