# OpenReview forum: "Attacking Few-Shot Classifiers with Adversarial Support Poisoning"
_ICML.cc/2021/Workshop/AML — ICML 2021 Workshop AML Poster_

### Official Review · Reviewer_PuSG · 2021-06-19
**Interesting Attack but Missing Some Details and Technical Novelty**

**Rating:** Reject
**Confidence:** 4

**Review:**

This paper introduces a poisoning-based attack against the few-shot classifiers. Specifically, it proposes to poison some samples in the support set with imperceptible perturbations. It also discusses how to conduct adversarial training based on the proposed attack.

The paper is well-written and the topic is of great significance and is suitable for this workshop. There are also lots of different types of experiments in the appendix. However, there are still lots of problems so that I cannot vote for acceptance.

Major Comments:
1.	Lacking technical novelty. Although the discussed task (i.e., poisoning few-shot learning) is new, the proposed attack and defense seem to be a simple extension of PGD and AT, respectively. At least the author should provide more details on why the proposed method works.
2.	The threat model is somehow unrealistic. Attackers not only need to know the model in (nearly) a white-box setting, they also need to have certain knowledge about the (seed) query set. I am not for sure whether this attack could actually happen in real-world scenarios. I suggest the author provides some real-world applications for more discussions or at least provides more discussion about the generalization of adopting the seed query set.
3.	I also have a concern about the experiments. It seems that the author only examined their attack on one few-shot classifier (i.e., ProtoNets). The author should adopt more different few-shot classifiers to verify their attack.


Minor Comments:

The discussion about the backdoor attacks in the appendix is very interesting. Their threat model seems to be more realistic. I would even suggest you do a separate research on it. However, because many readers may not be familiar with backdoor attacks, I suggest that the author can briefly introduce them or cite related surveys [1, 2].

References:

[1]  Data Security for Machine Learning: Data Poisoning, Backdoor Attacks, and Defenses. arXiv, 2020.

[2]  Backdoor Learning: A Survey. arXiv, 2020.

---

### Decision · Program_Chairs · 2021-06-21

**Decision:**

Accept (Poster)

**Comment:**

This paper introduced a poisoning attack against few-shot classifiers. The paper is well-written and the topic is suitable for this workshop. The reviewer raised several concerns about the novelty, threat model, and experiments of this paper. The authors are highly encouraged to address these concerns in the revised version.